# Existing Evidence from Economic Evaluations of Antimicrobial Resistance—A Systematic Literature Review

**DOI:** 10.3390/antibiotics14111072

**Published:** 2025-10-24

**Authors:** Sajan Gunarathna, Yongha Hwang, Jung-Seok Lee

**Affiliations:** International Vaccine Institute, Seoul 08826, Republic of Korea; sajan.gunarathna@ivi.int (S.G.); yongha.hwang@ivi.int (Y.H.)

**Keywords:** antimicrobial resistance, cost effectiveness, cost of illness, economic evaluations, limitations

## Abstract

**Background/Objectives**: Although antimicrobial resistance (AMR) is recognized as a critical global health threat across human, animal, and environmental domains, evidence from AMR economic evaluations remains limited. This study systematically reviewed available studies, emphasizing existing evidence and reported limitations in AMR-related economic evaluations. **Methods**: A comprehensive review of peer-reviewed empirical studies was conducted, including publications up to July 2023 without temporal restrictions, but limited to English-language articles. Literature searches were undertaken in PubMed and Cochrane using a search strategy centered on the terms “economic evaluations” and “antimicrobial resistance.” Screening and data extraction were performed by two reviewers independently, with disagreements resolved through consensus or consultation with a third reviewer. Findings were synthesized narratively. **Results**: Of the 3682 records screened, 93 studies were included. Evidence gaps were identified across income and geographic regions, particularly in low- and middle-income countries (LMICs) and the African, Southeast Asian, and Eastern Mediterranean regions. Studies were comparatively more numerous in high-income countries (HICs) and the European and Americas regions. Substantial gaps also existed in one health approach and community-based evaluations. Nine major study limitations were identified, with many interlinked. The most frequent issues included limited generalizability primarily due to inadequate sampling approaches (*n* = 16), and single-center studies (*n* = 11), alongside errors in cost estimation (*n* = 4), and lack of consideration for essential features or information (*n* = 3). **Conclusions**: The review highlights persistent evidence gaps and recurring methodological shortcomings in AMR economic evaluations. Addressing these limitations, particularly in LMICs, will strengthen the evidence base and better inform policy implementation to combat AMR effectively.

## 1. Background

Antimicrobial drugs, which include antibiotics, antivirals, antifungals, and antiparasitics, serve as essential medicines for preventing and treating infectious diseases in humans, animals, and plants [1]. Despite these agents’ life-saving impact over the years, antimicrobial resistance (AMR) is a significant global health challenge and is defined as the ability of microorganisms to counteract the action of antimicrobial agents, particularly when antibiotics lose their effectiveness in inhibiting bacterial growth [2]. The major drivers of AMR are excessive use and inappropriate prescription practices, including incorrect drug selection, duration, and frequency [3].

The World Health Organization (WHO) projected that the unrelenting overuse of antibiotics could result in 10 million global deaths by 2050 and may lead to 24 million people experiencing extreme poverty by 2030 [4]. Additionally, the World Bank (WB) estimated that AMR could result in USD 1 trillion in additional healthcare costs by 2050 and up to USD 3.4 trillion in gross domestic product losses per year by 2030 [5]. Furthermore, a total annual health sector cost of USD 28.2 million for methicillin-resistant *Staphylococcus aureus* in Canada was reported in 1998 [6], and a total annual hospital cost of USD 5.2 million for Ceftriaxone-resistant *Escherichia coli* bloodstream infections in Australia was reported in 2014 [7]. AMR has become a primary health concern globally because of its life-threatening impact and substantial economic burden. A recent systematic review reported that the global burden of AMR was due mainly to the pathogens *Escherichia coli*, *Acinetobacter baumannii*, *Klebsiella pneumoniae*, *Salmonella* spp. and *Staphylococcus aureus* [8]. This has elevated AMR to a critical global policy issue, and the current focus is to reduce its use, even though antibiotics are indispensable for human, animal, and plant health [9,10].

In this context, AMR evaluation studies are critical for addressing this policy issue by generating evidence on the burden of AMR and highlighting its severity. However, a review article highlighted that the prevailing studies are generally inadequate because of methodological gaps, including narrow perspectives of the analysis and problems associated with the input-cost data [11]. Good-quality research, including economic evaluations and comprehensive modeling, is needed to determine the most effective strategies to combat AMR and to find ways to address these methodological difficulties. More importantly, the number of economic evaluations of AMR is limited globally; moreover, existing evidence is disproportionately lower in LMICs.

With this background, it is critical to understand what evidence is currently available, and a review reporting the economic evidence and limitations of each published study of AMR-related economic evaluations would aid in designing and implementing policy interventions, ultimately helping combat AMR. Therefore, the main aim of this review is to report existing evidence and limitations in published AMR-related economic evaluations worldwide.

## 2. Methods

### 2.1. Study Design

We employed the methodology of systematic literature review, following the principles of the ‘Preferred Reporting Items for Systematic Reviews and Meta-Analyses’ (PRISMA) guidelines. The PRISMA Abstract Checklist and the full PRISMA Checklist are presented in Appendix A, respectively.

### 2.2. Eligibility Criteria

All peer-reviewed primary empirical studies on economic evaluations of AMR were assessed. There were no temporal restrictions for publications; studies published up to July 2023 were considered. The searches were limited only to the English-language publications.

### 2.3. Search Strategy

Data sources: The electronic search included bibliographic database searches via PubMed and Cochrane Library. The search was performed on 1 July 2023.

Search terms: The search strategy consisted of two high-level categories, namely, ‘economic evaluations’ and ‘antimicrobial resistance,’ which are searched via medical subject headings (MeSH) and text words. Each of the high-level categories included specific search terms, and the two high-level categories were combined as shown below:

[(“antimicrobial*” OR “antibiotic*” OR “antibacterial*” OR “antiviral*” OR “antifungal*” OR “antiparasitic*”) AND (“resistan*” OR “overus*” OR “misus*”) AND (“economic evaluation” OR “economic burden” OR “cost effective*” OR “cost utility” OR “cost benefit” OR “cost of illness” OR “cost minimization” OR “incremental cost-effectiveness ratio” OR “QALY*” OR “quality adjusted life year*” OR “DALY*” OR “disability adjusted life year*”)]

### 2.4. Selecting Studies

The outcomes of the searches in each database were exported and deduplicated using Mendeley Desktop 1.19.8 reference management software. Each study’s eligibility was determined based on the basis of the respective study title and abstract (and keywords where applicable) screening. Two independent reviewers assessed and further screened the full-text articles of the potentially eligible studies. A third reviewer resolved any disagreements regarding the inclusion of studies at both stages. The outcomes of this screening process are presented in a PRISMA flow diagram.

### 2.5. Data Extraction

We extracted data using a systematically developed form in Microsoft Excel that included all the necessary fields to achieve the review’s objectives. The data extraction table template is available in Appendix A.

The key indicators that we prioritized were infection-related data (type of infection, resistant pathogen, and intervention), economic evaluation-related information (type of economic evaluation, main outcome interest, and reported outcomes), and limitations/challenges reported in the studies. This review focused on six economic evaluation types, defined in the section ‘Definition of economic evaluations’ below. Two impartial reviewers meticulously extracted information on study attributes, economic assessments, and any limitations documented in the eligible research studies.

### 2.6. Definitions of Economic Evaluations

Cost-effectiveness analysis (CEA): CEA refers to a full economic evaluation that considers both the cost and effectiveness of each alternative intervention, and enables the combination of relevant outcome measures with costs, allowing alternatives to be ranked based on their effectiveness relative to resource utilization [12].

Cost-benefit analysis (CBA): CBA is a full economic evaluation that systematically compares the costs and benefits of an intervention to assess its economic profitability. In health economics, CBA compares the expected costs of healthcare interventions with their benefits to determine the most efficient use of resources. It is crucial for making well-informed decisions that enhance patient outcomes and optimize healthcare expenditures [13].

Cost-utility analysis (CUA): CUA is a full economic evaluation and that is considered an essential tool for evaluating and comparing the costs and effects of alternative interventions. CUA involves comparing the additional cost of a program from a specific perspective with the incremental health improvement expressed in terms of quality-adjusted life years (QALYs). CUA is a valuable tool for decision-makers aiming to maximize population health within budget constraints [14].

Cost minimization analysis (CMA): CMA is a full economic evaluation that evaluates and compares the costs of alternative interventions, helping to assess the value for money of an intervention. It is advantageous when both quantity (life years) and quality of life matter, as it measures health effects in terms of both by combining them into a single measure of health: QALYs. CMA is also a valuable tool for decision-makers aiming to maximize population health within a budget constraint [15].

Cost of illness (COI): COI is a partial economic evaluation that is commonly used to measure the economic burden of a specific disease or condition. It aims to identify and measure all costs associated with a particular disease, including direct, indirect, and intangible costs. Direct costs include medical costs, such as diagnostic tests, physician visits, medications, and nonmedical costs, including (but not limited to) travel expenses incurred when obtaining care. Indirect costs encompass productivity losses, such as work or leisure time missed due to the disease. Intangible costs are more challenging to quantify but may include factors such as reduced quality of life. The result of a cost of illness analysis is expressed in monetary terms, providing an estimate of the total economic burden of the disease on society. Decision-makers can use this information to understand the economic impact of a disease and prioritize interventions [16,17].

Disease burden studies: The burden of disease refers to the total and cumulative consequences of a specific disease or a range of harmful diseases within a community. These consequences encompass various aspects, including health impacts, social implications, and societal costs. The concept highlights the gap between an ideal scenario where everyone lives free of disease and disability and the current health status, thereby quantifying the overall burden. One widely used method to quantify the global disease burden is the measure of disability-adjusted life years (DALYs). These studies are crucial for understanding health priorities and informing public health interventions [18,19].

### 2.7. Quality Assessment

Considering the heterogeneity of the study designs of the selected studies, the Mixed-Methods Appraisal Tool (MMAT) was adapted for the quality appraisal of the eligible studies. The MMAT is a critical appraisal tool used in systematic reviews and meta-analyses that is designed to assess the methodological quality of included studies. It covers five study types: qualitative research, randomized and nonrandomized trials, quantitative descriptive studies, and mixed-methods studies [20]. The evaluation was performed by assigning a score of 1 if the criteria were fully met, 0.5 if the requirements were partially fulfilled, and a score of zero if the criteria were not met or if there was insufficient evidence to draw a conclusion. The sum of the scores for each study was then converted into percentages. A study with a score of more than 75.0% was classified as high quality, 50.0–74.0% as average quality, and less than 50.0% as poor quality. Two reviewers independently appraised the quality of each study, and a third reviewer was engaged when any disagreement occurred during the initial assessments.

### 2.8. Data Synthesis

An overview of the selected studies was presented, including the study citations, study year, country, income status according to the WB income classification, geographical distribution according to the WHO region classification, and type of economic evaluation. The primary outcome interests of this review, the available evidence and the study limitations of economic evaluations of AMR, were presented according to the type of economic evaluation.

## 3. Results

### 3.1. Search Results

A total of 3682 records were retrieved during the record identification stage by searching the electronic databases PubMed (*n* = 2568) and Cochrane (*n* = 1114) (Table 1). After the title, abstract, and full-text screening, 93 articles met the criteria for inclusion in the review (Figure 1).

### 3.2. Quality Appraisal of Eligible Studies

Among the five study designs considered in the MMAT, this review included only three types of studies, including quantitative randomized controlled trials (*n* = 30, 32.3%), quantitative nonrandomized studies (*n* = 29, 31.2%), and quantitative descriptive studies (*n* = 34, 36.5%).

Among the quantitative randomized controlled trials, the majority (*n* = 24, 80.0%) were high-quality studies, five (16.7%) trials were of average quality, and one (3.3%) was rated as poor quality. However, among the quantitative nonrandomized studies, most of the studies (*n* = 15, 51.7%) were average-quality studies, followed by 12 (41.4%) high-quality studies and two (6.9%) poor-quality studies. Most of the quantitative descriptive studies were also high-quality studies (*n* = 21, 61.8%), followed by 10 (29.4%) average-quality studies and three (8.8%) poor-quality studies.

Overall, this review included 57 (61.3%) high-quality studies, 30 (32.3%) average-quality studies and six (6.5%) poor-quality studies (Appendix A).

### 3.3. Overview of AMR Economic Evaluations

Among the selected studies (*n* = 93), 25 (26.2%) either did not report the type of economic evaluation or reported in a way that did not match our classification criteria. Therefore, we reclassified these into six types of economic evaluations (CEA, COI, CBA, CUA, CMA, or disease burden studies) based on their primary outcome of interest. The original and revised classifications are presented in Appendix A.

Most studies were single-country analyses (*n* = 84, 90.3%) while 8 (8.6%) involved multiple countries, and one (1.1%) study did not specify the study location. Considering both single- and multiple country studies (*n* = 93), the total amounted to 162 distinct evaluations conducted across 58 countries.

Considering the geographical distribution, Figure 2 presents the spread of AMR economic evaluations across regions based on the WHO regional classification. Across all 162 analyses from 93 studies in 58 countries, the European region accounted for the largest share (52.5%, 85 analysis across 34 countries), followed by the Americas (21.6%, 35 analysis across 4 countries), the Western Pacific region (17.9%, 29 analysis across 10 countries), the African region (4.3%, 7 analysis across 7 countries), the Eastern Mediterranean region (1.9%, 3 analysis across 3 countries), and the South-East Asia region (1.9%, 3 analysis across 3 countries) (Figure 2). Moreover, overview of the studies, including country and income classifications, is provided in Appendix A.

Most of the studies were CEAs (*n* = 50, 53.7%), followed by COIs (*n* = 33, 35.4%), disease burden studies (*n* = 5, 5.4%), CBAs (*n* = 2, 2.2%), CUAs (*n* = 2, 2.2%), and a CMA (*n* = 1, 1.1%). The distribution of economic evaluations according to the WB income classification of countries is presented in Table 2.

### 3.4. Available Evidence of Economic Evaluations of AMR

All the studies were human-centered studies, and no animal- or environmental-related studies were found during this review. With respect to the study setting of all eligible studies, most studies were hospital-based (*n* = 74, 79.6%), including tertiary-care hospital-based studies (*n* = 9, 9.7%), followed by community-based (*n* = 8, 8.6%) and combined hospital- and community-based (*n* = 4, 4.3%) studies. Furthermore, seven studies (7.5%) did not reveal information related to the study setting. All the human center economic evaluations of AMR involved various infections, interventions, and resistant pathogens, which are presented according to the value type, unit of measurement and reported values in Appendix B.

### 3.5. Limitations Reported in Studies of AMR-Related Economic Evaluations

#### 3.5.1. Cost-Effectiveness Analysis

Among the 50 CEAs, the most common limitations reported were overestimation of results [21,22,23,24], underestimation of results [23,25,26,27,28,29,30,31], absence of primary data [21,28,32,33,34,35,36,37,38,39,40], lack of consideration of essential features or information [21,26,28,31,32,35,38,41,42,43,44,45,46,47,48,49,50], uncertainty [25,31,34,39,44,51,52,53], inadequate sampling approaches [21,22,29,35,41,44,54,55], assumption errors [23,30,32,47,55,56,57,58], errors in cost estimation [25,26,31,35,42,44,45,49,56,59,60,61,62,63,64], lack of generalizability/standardization of study results to the population [22,26,29,34,35,42,43,44,45,53,62], and other limitations [24,36,48,55,57,58,63]. More details are shown in Table 3. Three CEA studies out of 47 studies did not report limitations.

#### 3.5.2. Cost of Illness Studies

Among the 33 COI studies, the most common limitations reported were the underestimation of results [65,66], absence of primary data [66,67,68,69], no consideration of essential features or information [70,71,72,73,74], uncertainty [75,76], inadequate sampling approaches [6,66,73,77,78,79,80], assumption errors [75], errors in cost estimation [6,7,67,68,70,71,72,76,78,81,82,83,84,85,86], lack of generalizability/standardization of study results to the population [66,67,69,70,73,79,81,84,85,86,87,88], and other limitations [65,68,71,73,77,79,89], as shown in Table 4. However, study limitations were not reported in one COI study.

#### 3.5.3. Other Economic Evaluations: Disease Burden Studies, Cost-Benefit Analysis, Cost-Utility Analysis, and Cost Minimization Analysis

Among the disease burden studies, the absence of primary data [90,91,92,93] and the lack of consideration of essential features or data/information [90,93,94] were the significant limitations reported. The lack of generalizability/standardization of study results to the population was the most common limitation noted in the cost-benefit analysis [95,96], cost-utility analysis [97,98], and cost minimization analysis [99]. Further details are presented in Table 5.

### 3.6. Causes and Consequences of Study Limitations

Figure 3 shows the interactions of limitations, showing how one limitation leads to another. The most common limitation interactions were the lack of generalizability/standardization of the study results to the population due to inadequate sampling approaches (reported in 16 studies) [6,21,29,35,41,42,44,54,55,66,67,77,78,79,92,99], single-center studies (reported in 11 studies) [22,41,44,55,73,78,79,80,84,96,98], errors in cost estimation (reported in 4 studies) [70,73,81,87], and no consideration of essential features or information [69,85,88] (reported in 3 studies). Additionally, the absence of primary data led to errors in cost estimation in 8 studies [35,40,45,47,52,60,67,85] and to the uncertainty of the study findings in 7 studies [21,25,31,34,39,52,53]. Furthermore, errors in cost estimation led to underestimations of results in 6 studies [25,27,28,29,31,66]. Further interactions are also presented in Figure 3.

## 4. Discussion

We conducted a systematic review of published literature up to July 2023, to synthesize all available evidence from economic evaluations related to AMR. Among the 3682 records identified, 93 eligible articles were included in our review. These articles included 50 CEAs, 33 COIs, five disease burden studies, two CBAs, two CUAs, and one CMA.

### 4.1. Available Evidence and the Evidence Gap in AMR Economic Evaluations

Nearly 75% of the studies originated from HICs, highlighting the disparity in evidence across varying income contexts. The absence of sufficient evidence in LMICs is particularly concerning, given that patients in these regions may require more potent medications, prolonged hospital stays, and additional medical interventions to treat AMR-related diseases. In resource-constrained settings, managing antibiotic-resistant infections poses greater challenges and incurs higher costs [100]. Conducting economic evaluations related to AMR can aid in allocating resources effectively in such resource-limited contexts. The significance of evidence in LMICs becomes apparent when considering the substantial 76% increase in antibiotic usage between 2000 and 2018, while HICs maintained relatively stable consumption during the same period [101]. Furthermore, in contrast to HICs, LMICs and LICs exhibit higher rates of antibiotic misuse due to overuse and self-prescription [102]. Given these challenges, strengthening the global evidence concerning AMR in LMICs is imperative.

Studies are more abundant in the region of Americas, the Western Pacific and European regions. However, only 2–3 articles were published for each African, Southeast Asian, and Eastern Mediterranean region. This pattern suggests that while numerous studies originate from HICs, they predominantly focus on the Americas, Western Pacific, and European regions rather than HICs in other regions. Nevertheless, expanding the evidence base for economic evaluations in other regions is crucial. A recent systematic review revealed that countries in the African region reported 18% higher rates of antibiotic misuse than other nations did [102]. Additionally, the World Health Organization has highlighted the substantial challenges faced by the African region due to inadequate enforcement and rapid proliferation of antibiotic-resistant strains resulting from misuse and overuse of antibiotics. These issues pose a significant threat to Africa’s public health and healthcare system, leading to increased morbidity, mortality, and healthcare expenses [103,104,105,106]. Furthermore, the Southeast Asia region faces significant challenges, as it is at risk of the emergence and spread of AMR in humans. Factors contributing to this risk include poverty, inadequate sanitation, and the overuse of antibiotics [107]. Moreover, a study conducted across 139 hospitals in seven Eastern Mediterranean countries revealed high overall hospital antimicrobial usage [108]. Considering the above findings, strengthening the global evidence concerning AMR in other regions where evidence is lacking is imperative.

Furthermore, despite the high importance and necessity of the One Health Approach for understanding and mitigating AMR, all the available economic evaluations have focused on human health. No studies were found in the Animal or Environmental related fields. This represents a limitation of the one health approach, which aims to sustainably balance the health of people, animals, and ecosystems [109]. A recent systematic review emphasized the importance of one health approach in AMR research, particularly for informing AMR policy decisions [110]. However, achieving these goals remains uncertain due to the lack of evidence related to animal and environmental health.

Additionally, community-based health studies play a crucial role in contextual learning within real-world settings, allowing us to understand health issues, observe health behaviors, and explore social determinants [111]. Unfortunately, the review revealed that most of the studies were conducted in hospital settings, leaving a gap in evidence for community-based evidence.

### 4.2. Limitations Reported in AMR Economic Evaluations and Their Interactions

We identified nine significant limitations associated with AMR economic evaluations: overestimation of results, underestimation of results, absence of primary data, no consideration of essential features or information, uncertainty, inadequate sampling approaches, assumption errors, errors in cost estimation, lack of generalizability/standardization of study results to the population, etc. When reviewing the limitations of AMR economic evaluations, we found that one limitation often leads to another, creating interconnected links among these limitations. The most common interaction occurs with the lack of generalizability/standardization of study results to the population. This limitation is influenced by other limitations, such as inadequate sampling approaches, reliance on single-center studies, errors in cost estimation, and no consideration of essential features or information.

Inadequate sampling approaches occur because a sample may not exhibit precisely the same behavior as the larger population from which it was selected. It may either underrepresent or overrepresent the study population, especially due to the non-random nature of the sample. For these reasons, inadequate sampling approaches are often recognized as one of the major reasons for the uncertainty and limited generalizability of the study findings [112,113]. On the other hand, single-center studies (a type of sampling error), which are conducted in a single hospital, clinic, or research institution, often involve a homogeneous sample due to limited diversity, as all data collection, participant recruitment, and intervention occur within this location. Therefore, the findings of these studies are generated under context-specific factors and geographical bias, which ultimately affect their generalizability [114,115].

Moreover, we identified the presence of a lack of applicability, and the uncertainty of the study results due to errors in cost estimations [25,27,28,29,31,66]. Errors in the data analysis could affect the applicability of the study findings, as inaccurate measurements or misclassification undermine the validity of the study results. According to this review, one of the reasons for errors in cost estimations is the absence of primary data required for the analysis, which is an uncontrollable limitation, as evidenced by 14 studies [21,25,31,34,35,39,40,45,47,52,53,60,67,85]. Furthermore, the lack of consideration of essential data/information, which could be due to a lack of data/information, also affects the generalizability and standardization of the study results to the population [69,85,88].

Consequently, considering the potential drawbacks and limitations, several critical factors should receive greater attention while addressing existing gaps in evidence. These factors include sample characteristics (such as the representativeness of the sample), the study context (for example, whether it occurs in a controlled laboratory environment), the relevance of the study over time (given societal, technological, and environmental changes), measurement tools, research design, and cultural and geographical considerations, which could impact the generalizability or standardization of study outcomes [113,116,117].

## 5. Strengths and Limitations

The key advantage of this review lies in our utilization of a comprehensive and sensitive search strategy to capture all published articles related to economic evaluations of AMR. Additionally, we deliberately avoided restricting the search time frame, ensuring that all relevant publications, regardless of their publication date, were included in our analysis up to July 2023.

However, several limitations are associated with this review. First, our search was confined to two research databases. Second, we did not consult the reference lists of eligible studies. Third, we focused exclusively on peer-reviewed journal articles, excluding gray literature, which might have restricted the inclusion of relevant information from published or unpublished sources. Fourth, our search was limited to English-language content, potentially introducing language bias. Fifth, we excluded articles that were not available online and did not contact corresponding authors for access to relevant publications. Finally, we categorized other types of economic evaluations into six predefined categories, which could introduce misclassification bias.

## 6. Conclusions and Recommendations

This review sheds light on the existing evidence gaps in LMICs, with a particular focus on the African, Southeast Asian, and Eastern Mediterranean regions. These gaps extend to the community-based approach, whereas the One Health Approach suffers from a complete lack of studies related to animal and environmental aspects. Additionally, the review identifies a chain of interrelated limitations, where one limitation often leads to another. The most common limitation lies in the lack of generalizability or standardization of the study outcomes to the population.

It is imperative that future studies address the identified evidence gaps more effectively. Specifically, researchers must focus on tackling the root causes of the most common limitations. These limitations, if left unaddressed, could significantly impact the validity of the study findings. Moreover, addressing these gaps would contribute to evidence-informed policy development aimed at combating AMR.

## Figures and Tables

**Figure 1 antibiotics-14-01072-f001:**
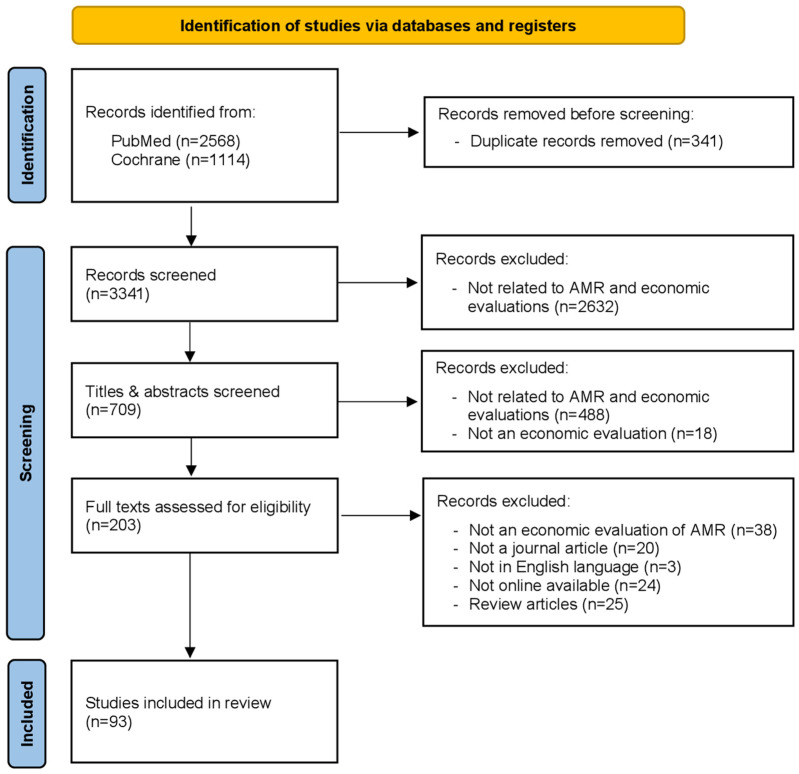
PRISMA flow diagram.

**Figure 2 antibiotics-14-01072-f002:**
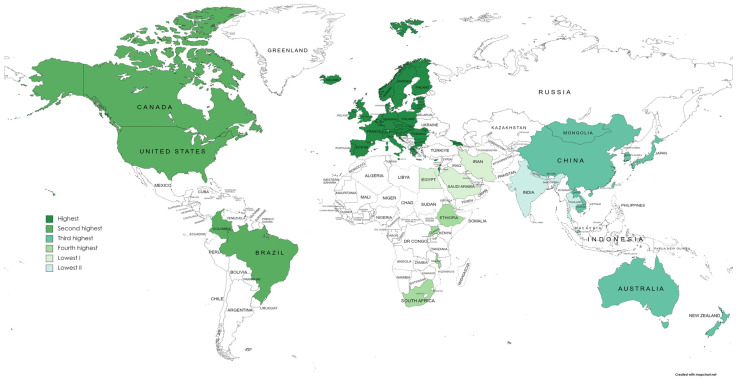
Countries reported AMR economic evaluations according to WHO region classification. Note: **Highest**: 85 analyses (52.5%) in 34 European countries (the UK, Netherlands, Germany, France, Greece, Belgium, Italy, Spain, Switzerland, Austria, Poland, Portugal, Slovakia, Slovenia, Bulgaria, Croatia, Cyprus, Czech Republic, Denmark, Estonia, Finland, Hungary, Iceland, Ireland, Israel, Latvia, Lithuania, Luxembourg, Malta, Norway, Romania, and Georgia, and Moldova); **second highest**: 35 analyses (21.6%) in four regions of the Americas (USA, Canada, Brazil, and Colombia); **third highest**: 29 analyses (17.9%) in 10 Western Pacific countries (Japan, Australia, Singapore, Republic of Korea, New Zealand, Taiwan, Cambodia, Lao PDR, Mongolia, and China); **fourth highest**: seven analyses (4.3%) in seven African countries (Malawi, Ethiopia, Uganda, and South Africa); **lowest I**: three analyses (1.9%) in three Eastern Mediterranean countries (Saudi Arabia, Egypt, and Iran); **lowest II**: three analyses (1.9%) in three South East Asian countries.

**Figure 3 antibiotics-14-01072-f003:**
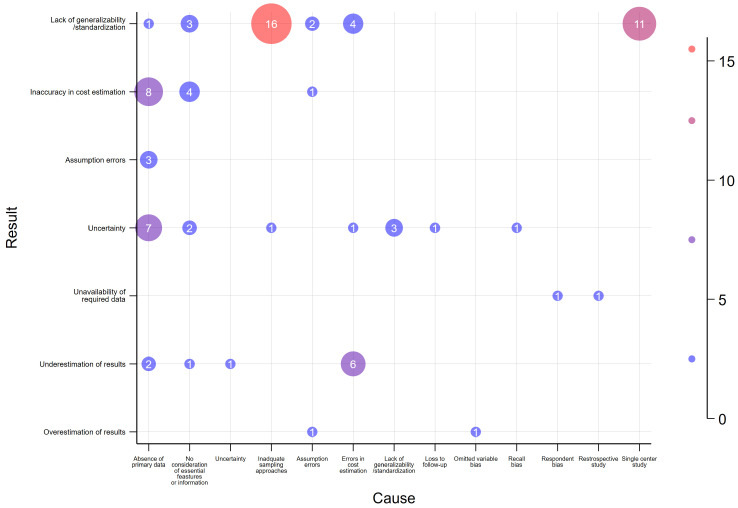
Causes and consequences of study limitations.

**Table 1 antibiotics-14-01072-t001:** Database-specific search strategy and number of results per database.

Database	Search Strategy	Number of Records
PubMed	(“antimicrobial*”[Title/Abstract] OR “antibiotic*”[Title/Abstract] OR “antibacterial*”[Title/Abstract] OR “antiviral*”[Title/Abstract] OR “antifungal*”[Title/Abstract] OR “antiparasitic*”[Title/Abstract]) AND (“resistan*”[Title/Abstract] OR “overus*”[Title/Abstract] OR “misus*”[Title/Abstract]) AND (“economic evaluation”[Title/Abstract] OR “economic burden”[Title/Abstract] OR “cost effective*”[Title/Abstract] OR “cost utility”[Title/Abstract] OR “cost benefit”[Title/Abstract] OR “cost of illness”[Title/Abstract] OR “cost minimization”[Title/Abstract] OR “incremental cost-effectiveness ratio”[Title/Abstract] OR “qaly*”[Title/Abstract] OR “quality adjusted life year*”[Title/Abstract] OR “daly*”[Title/Abstract] OR “disability adjusted life year*”[Title/Abstract])	2568
Cochrane	((Antimicrobial*):ti,ab,kw OR (Antibiotic*):ti,ab,kw OR (Antibacterial*):ti,ab,kw OR (Drug*):ti,ab,kw OR (Medicine*):ti,ab,kw OR (Antiviral*):ti,ab,kw OR (Antifungal*):ti,ab,kw OR (Antiparasitic*):ti,ab,kw) AND ((Resistan*):ti,ab,kw OR (Overus*):ti,ab,kw OR ((Over NEXT usage*)):ti,ab,kw OR (Misus*):ti,ab,kw) AND ((“Economic evaluation”):ti,ab,kw OR (“Economic burden”):ti,ab,kw OR ((Cost NEXT effective*)):ti,ab,kw OR (“Cost utility”):ti,ab,kw) AND ((“Cost benefit”):ti,ab,kw OR (“Cost of illness”):ti,ab,kw OR (“Cost minimization”):ti,ab,kw OR (“Incremental cost-effectiveness ratio”):ti,ab,kw OR (QALY*):ti,ab,kw AND ((Quality adjusted life NEXT year*)):ti,ab,kw OR (DALY*):ti,ab,kw OR ((Disability adjusted life NEXT year*)):ti,ab,kw)	1114
Total records		3682

**Table 2 antibiotics-14-01072-t002:** AMR economic evaluations according to the WB income classification.

Income Level	CEA		COI		Other Economic Evaluations **		Total *	
	*n*	%	*n*	%	*n*	%	*n*	%
High-income countries	39	60.0	22	33.8	4	6.2	65	100
Upper middle-income countries	8	57.1	5	35.7	1	7.2	14	100
Lower middle-income countries	0	0	3	100	0	0	3	100
Low-income countries	1	50.0	1	50.0	0	0	2	100
Total *	48	57.1	31	36.9	5	6.0	84	100

Note: * Only single country studies were included in this classification. ** CBA, CUA, and CMA studies were included in other economic evaluations.

**Table 3 antibiotics-14-01072-t003:** Limitations associated with AMR-related CEA.

Reported Limitations	Description/Cause(s) of the Reported Limitation
Overestimation of results	Patients initially treated with ertapenem (treatment I) are less likely to show success with imipenem (treatment II) because it is in the same class as ertapenem. As a result, the QALYs gained with ertapenem were arguably overestimated [21]. Accurately estimating the incremental length of hospital stay attributable to *S.aureus* bacteremia is affected by omitted variables and simultaneity biases. Therefore, the estimations might be relatively high [22]. The effectiveness of genotypic antiretroviral resistance testing may have been overestimated if failure to prescribe optimal therapy [23]. Overestimation of total cost due to the unrealistic assumption; all antibiotic treatment days are associated with nosocomial pneumonia [24].
Underestimation of results	Only included production loss for working-aged people (20–64 years). This probably led to an underestimation of the true cost-effectiveness [25]. The model used does not capture the difference in resistance profile between two antibiotic groups, which resulted in an underestimation of the cost and QALY advantages observed over time [30]. The effectiveness of genotypic antiretroviral resistance testing may have been underestimated if, over time, clinicians learn to improve their choice of GART-guided therapies [23]. The difference in benefits between the two gonorrhea treatment strategies may have been underestimated [26]. Hospital length of stay was likely underestimated as healthcare resource utilization and cost data in the clinical trial were only measured through the end of the study visit for all modified intent to treat patients [27]. Underestimating incremental costs as suspected cases might be included in the cohort [28]. Underestimation of length of stay as measured healthcare resource use and costs through the end of study visits [29]. Underestimation of hospitalization costs as they were not adjusted for inflation over the years [31].
Absence of primary data	Only diabetic-specific survival estimates were used for patients with sequelae [21]. The study had to rely on the UK cost-effectiveness threshold as the unavailability of the European-wide threshold [32]. The published data on the utility of the various health states examined in the model is unavailable [33]. Lack of information on long-term survival and optimal duration of nivolumab [34]. The model utilized published data from other populations due to the unavailability of data on the long-term mortality and the health utility of patients with carbapenem-resistant *Klebsiella pneumoniae* bloodstream infection [35]. The absence of a specific ICD-9-CM diagnosis code for Nosocomial Pneumonia and our reliance on an algorithm to exclude patients thought to have community-acquired pneumonia [36]. Due to the unavailability of used data on the prevalence of NS5A resistance from the European and USA population, they may not reflect the prevalence in the UK population [37]. Lack of reported data on the relationship between resistance and clinical failure/the model requires using population-specific efficacy data, and the meta-analyses found in the literature were unsuitable [38]. High amount of missing data [39]. Unable to directly assess esketamine’s cost-effectiveness versus alternatives due to the absence of comparative outcomes data [40]. Data sources for this study were unavailable daily, which could lead to time-dependent bias [28].
No consideration of essential features or information	Did not subgroup chronic gastritis patients into nonatrophy, atrophy, and intestinal metaplasia groups, which are other potential factors that might affect *H. pylori* eradication [41]. Only diabetic-specific survival estimates were considered for patients with sequelae/adverse events [21]. The intervention may not be compared with the most relevant alternatives as this is a placebo-controlled trial [32]. The decision analytic model focused on mild to moderate Community-acquired pneumonia (CAP) but did not consider severe CAP [42]. Did not consider gonorrhea-positive patients coinfected with another organism, which would affect patient pathways and treatment options [43]. The Markov model did not consider the relapse rates and rehospitalizations [35]. A budget impact analysis was not performed [44,45]. The model excludes several significant factors, including the potential benefits to the PTSD patients’ families, the broad reduced risks, domestic violence, severity of substance abuse and the criminal justice system’s involvement [46]. The value of the intervention in which patients shared rooms has not been explored [47]. Treatment-related adverse effects were not addressed [38]. Did not account for different anatomic locations of gonorrhea infections and the import of gonorrhea infections with resistant strains [26]. Did not evaluate the cost-effectiveness of testing for other resistance variants (e.g., NS5B and NS3) and the performance characteristics (e.g., sensitivity, specificity) of a diagnostic test [45]. Suspected cases and other hospital-acquired and multidrug-resistant infections were not actively excluded from the cohort [28]. Did not specifically compare the periods before or after the diagnosis of infection [48]. The analysis of the feasibility of obtaining the AMR reductions has not been validated [49]. The analysis did not capture the broader effects of hearing loss on the ability to work [50]. Possible differences in future populations, demographic changes (including the growth in the number of older people)/The constraints of the model led to the use of a limited number of pathogens (e.g., three pathogens in this study)/The diseases and pathogens analyzed were limited [31].
Uncertainty	Uncertainty of input data. Uncertainty of input data due to sampling errors and lack of data [21]. Not all the participants completed all the parts of the questionnaire/recall bias as collecting data over six months from the patients with depression [39]. Data were generated not from the fundamental analyses but from the established two-parameter Weibull survival model [51]. The available data was that survival time was censored on the date of progression for patients who had not died [52]. The vaccine protection duration is not fully understood and is uncertain [53]. Uncertainty of the results. Uncertainty about the results as input values and transmission probabilities were collected from different studies and combined in this study [25]. Uncertainty of the results due to mis-estimation influence of used data obtained from other sources [34]. The impact from a societal perspective is unclear due to the perspective taken in the analysis. The accuracy of cost-effectiveness may be restricted due to using the EQ-5D-3 L to assess the health-related utility [44]. The absence of empirical research led to reliance on expert opinion, which made the uncertainty of the estimates of hospital length of stay [31].
Inadequate sampling approaches	Errors in the sampling method [21]. Small sample size—sample of 180 [41], sample of 263 [54], sample of 3000 [35], sample of 96 [55], sample of 1225 [29]. Single-center study [22,41,44,55].
Assumptions errors	Using false/unrealistic assumptions. Assuming that the cost of resistance is the same across all antibiotics regardless of antibiotic class and the sector of care/assuming that there is a linear relationship between prescribing and resistance; however, in practice, the impact will be lagged, and there may be a nonlinear relationship [32]. Assuming patients received all required/necessary treatments; however, patients may not receive all for various reasons, including disease progression and personal choice [56]. Antibiotics used to treat other Gram-positive and Gram-negative pathogens were assumed to be similar between both groups [55]. Assumed a general medicine ward set up with 20 single bedrooms, which may not be the configuration of all general medicine wards [47]. The assumption is that patients in the treatment completion state have the same health characteristics as the general population [57]. The use of resources in the clinical trial may not represent the actual clinical practice since clinical trials are conducted in a selected population that meets the inclusion and exclusion criteria [58]. Using simplification compared to routine practice: all patients are treated with piperacillin/tazobactam, or all patients are treated with ertapenem [30]. Using unproven assumptions about the mechanism of disease progression [23]. Using a higher number of assumptions [57].
Errors in cost estimation	Direct cost. limited to the cost of pharmaceutical treatments and the administration of those treatments, not taking into consideration the cost of treating other sequelae of castration-resistant prostate cancer, such as pain, spinal cord compression, or palliative care [56]. Only productivity loss was included, direct cost was missing [25]. Costs associated with prolonged hospitalization due to nephrotoxicity were ignored [35]. Direct nonmedical cost was not included [26,59]. The cost of monitoring vancomycin plasma levels is not included, which is necessary to obtain optimal results with this treatment/treatment costs of adverse reaction outcomes for both products, which could influence the global decision-making process [60]. The cost of treatment-related adverse events was not accounted for in the analysis [45]. Hospitalization costs have not been adjusted for inflation over the ten years [31]. Indirect cost. Did not consider indirect costs [35,59,61,62]. Costs of productivity loss were not considered [42,63,64]. The cost of welfare services was not included [44]. Not considering the additional costs associated with human and health resources of implementing approaches/future health costs associated with improved survival are omitted [49].
Lack of generalizability/standardization of study results to the population	Study results cannot be extrapolated to the patients with severe community-acquired pneumonia, only for the moderate patient population [42]. Lack of generalizability and applicability due to AMR rates constantly changing/generalizable only to England, not other countries [43]. The true cost-effectiveness could differ from the study findings as real-life survival or nivolumab use varies substantially from the values in this study [34]. Findings cannot be generalized for other Chinese provinces and countries due to significant variations in healthcare resources and epidemiology of *K. pneumoniae* resistance [35]. Cannot be generalizable to other populations as the participants may not represent the typical background of depressed patients in Japan, only for hospitalized patients [44]. Cannot be fully generalizable to hospitals with different care levels or settings in other countries [22]. Study results were restricted to men and the health benefits in men; only the population of men who had sex with men [26]. Generalizability of the results for other races/ethnicities or in other countries may be limited due to the heterogeneity of payer perspectives and the country-specific epidemiologic data used [45]. It is generalizable to other settings where only introducing drug-resistant strains may lead to prolonged outbreaks of typhoid fever [53]. Generalizable only to the population of the study setting, and the applicability of this data to other patients is also limited [62]. The vancomycin dosing used in the study may not reflect real-world dosing practice patterns [29].
Others	Lost to follow-up [57]. Retrospective study [48,55,58,63]. Using proxy estimates of costs due to the unavailability of claims data [36]. Respondent bias as using the Delphi method [24].

**Table 4 antibiotics-14-01072-t004:** Limitations associated with AMR-related COI.

Reported Limitations	Description/Cause(s) of the Reported Limitation
Underestimation of results	The results were underestimated due to difficulties distinguishing between infection and colonization [65]. The results were underestimated due to underestimating severe pneumonia patients and medical costs [66]. The results were underestimated due to the changing behavior of the clinicians to fit the expected results in the observational study—Hawthorne effect [79].
Absence of primary data	Lack of availability of usable estimates of the bed day cost [67]. No related prospective trials were available for the similar patient population [68]. More data was missing from the study participants [69]. The database did not report laboratory-based information including microscopy on the infecting pathogens [66].
No consideration of essential features or information	Not possible to estimate the impact of adverse events onward costs [70]. Cannot distinguish colonization or infection, which incur significantly different costs due to the nature of the retrospective study [71]. Social burden due to deterioration of quality of life from illnesses was not considered/In the process of selecting matching patients for the control groups, some patients were not selected for the control groups and were excluded from the analyses [72]. Not including several essential patient subgroups [73]. Not including antibiotic exposure data [74].
Uncertainty	Uncertainty of the input data. Direct and indirect costs for this study were derived from the published literature and may not fully represent real-world costs [75]. Data sources depend on the coding quality of hospital records and do not reflect the actual costs to the hospital [76]. Uncertainty of the study results. Results were uncertain as the analysis is based on the population included in Phase III clinical trials and may not represent the overall TRD population/all patients were required to take their medication under medical supervision, which does not reflect real-world practice [75].
Inadequate sampling approaches	Small sample size—sample of 34 [77], sample of 99 [6], sample of 1539 [78]. Limited to the sample of one-region/single-center study [73,78,79,80]. Selection bias in MRSA cases was identified using anti-MRSA drugs [66].
Assumptions errors	Total employment (100%) is assumed when estimating the indirect cost [75]. All patients were assumed to follow a similar and consistent pattern of initiating a new line of therapy following a relapse, which may not represent the actual treatment among these patients [75].
Errors in cost estimation	Direct costs. Cost estimates may not reflect costs incurred at specific institutions [70]. The cost of multiple infections or pathogens was excluded [76]. Not considered all direct medical costs—only considered the excess cost to treat bloodstream infection [67], only considered the laboratory investigation cost [81]. Direct nonmedical costs were excluded [71,82]. Additional costs, such as investigations, follow-up outpatient visits, and management costs, were excluded [6,78,83]. Hospitalization costs were not separated into pre- and postdiagnosis of the infection [84]. Costs associated with medical equipment, staff time, and other additional aspects of treating a patient in the hospital were omitted [7]. Data were extracted from an administrative claims database; therefore, the findings are subject to potential miscoding [85]. Indirect costs. Not all the indirect costs are accurately estimated/included [81]. Excluded indirect costs [68,71,72,78,82,86].
Lack of generalizability/standardization of study results to the population	Lack of generalizability as the cost values were not representative of the whole region [70,81,87]. Results were based on acute care hospital data, which may not apply to other healthcare settings [67]. Lack of generalizability to other regions as the study took place only in South Texas [79]. The generalizability issue was due to the experience of trial participants, which was different from that of patients seen in routine practice [69]. The model built under which the clinical trial was performed may differ from real-life clinical practice in Germany [86]. The implications are only valid for hospital-based, as the impact of AMR in the community has not been included [88]. Lack of generalizability due to the samples may not represent all severe community-onset pneumonia patients [66]. The study involved only two institutions, and the findings may not be generalizable to other institutions [84]. The costs used to estimate lost productivity from hospitalization and death were national averages and may not apply to a sicker population/Costs and mortality rates were measured in a subset of hospital patients who were at high risk and suffering with severity of illness; therefore, results cannot be applied to all patients in the community [73]. The treatment failure algorithm used to identify patients with treatment-resistant depression (TRD) may not be representative of all patients with TRD [85].
Others	Retrospective study [65,68,71,73,79,89]. Methodological errors. Healthcare facilities were not randomly selected [77]. Patients were not blinded [79].

**Table 5 antibiotics-14-01072-t005:** Limitations associated with other economic evaluations of AMR.

Type of Economic Evaluation	Reported Limitations	Description/Cause(s) of the Reported Limitation
Disease burden (*n* = 5)	Absence of primary data	The unavailability of national surveillance data in Greece compelled the reliance entirely on data from the only existing national point prevalence survey data [90]. Many of the parameters required for estimating DALYs were borrowed from previous studies/data about the length of stay in Japanese hospitals, which were scarce [91]. In the absence of Australian estimates of morbidity and mortality attributable to AMR, used estimates from Queensland to project to the Australian population [92]. Published data on clinical responses to treatments is considered lacking [93].
	No consideration of essential features or information	The strength of the evidence supporting each parameter estimate was not graded based on the clinical outcome studies’ statistical analysis method/Did not adjust the models for age-specific risks, coinfections, appropriateness of antibiotic therapy, or type of care [94]. The analysis did not account for treatment factors [90]. Drug allergies or adverse events were not considered in the model [93].
	Others	Underestimation of results—Underestimation due to the exclusion of patients due to missing data [91]. Uncertainty of input data—The input data were uncertain as the data for the disease models were retrieved from systematic literature reviews, which were varied in the representativeness of required evidence, availability and quality [94]. Assumption errors-Assumed there are common transition probabilities for all subgroups [94]. Errors in cost estimates—did not consider societal cost, which covers a range of health services, health infrastructure, and disease prevention [92]. Lack of generalizability/applicability of study results to the population—Used data limited in representing the entire Australian population [92].
CBA (*n* = 2)	Lack of generalizability/standardization of study results to the population	Lack of generalizability to other settings [95]. Data on unit costs were mainly derived from the university hospital and thus may not be comparable to those at other hospitals [96].
	Others	Absence of primary data—unavailability of molecular typing necessary for the analysis [95]. Assumption errors—assumption of no differences in patient contacts or care due to the use of gowns [95]. Errors in cost estimation—Omission of blood culture costs [95].
Cost–utility analysis (*n* = 2)	Lack of generalizability/standardization of study results to the population	Applicable only to a patient population with similar characteristics to those included in this study [97]. The information may not apply to patients with newly developed healthcare-associated urinary tract infections, and the generalization of these results in different clinical settings with less severe patients should be limited [98].
	Others	Absence of primary data—The long-term quality of life and survival were extrapolated from the literature [97]. No consideration of essential features or information—Did not account for several factors, such as the emergence of drug resistance during treatment, the possibility of antibiotic-related adverse reactions, and the risk of other healthcare-associated infections during hospitalization [98]. Single-center study [98].
CMA (*n* = 1)		Inadequate sampling approaches—lower sample size (*n* = 50) [99]. Lack of generalizability/standardization of study results to the population—results did not apply to other healthcare systems [99].

## Data Availability

All the data are included in the manuscript and in the public domain.

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
