# Peer review of "Existing Evidence from Economic Evaluations of Antimicrobial Resistance—A Systematic Literature Review"

_antibiotics, 2025, doi:10.3390/antibiotics14111072_

Round 1
Reviewer 1 Report
Comments and Suggestions for Authors
The document presents a well-structured and developed systematic review, with a level of detail and precision considered relevant for this type of writing. The topic is pertinent given the current situation of antimicrobial resistance, a growing problem with global impact. Therefore, it is essential to have solid and well-documented economic evaluations that enable the prioritization of interventions and the efficient allocation of resources, as well as the design of programs that mitigate the health and economic impacts.
The paper rigorously analyzes the available literature on economic evaluations, clearly describing the search strategy, selection, and critical analysis of the included studies. I have only a few observations. First, it is recommended that Figure 2 be reviewed and corrected, as it appears incomplete in the current document, making it difficult to correctly interpret the information it seeks to present. On line 233, the country Columbia is mentioned; however, the correct name is Colombia. The same is true for supplementary materials 6 and 7. It is also recommended to write the name of the microorganisms in italics throughout the document, especially in Table 3, with the first letter of the first name capitalized, for example: Staphylococcus aureus.
Author Response
Comment 1: The document presents a well-structured and developed systematic review, with a level of detail and precision considered relevant for this type of writing. The topic is pertinent given the current situation of antimicrobial resistance, a growing problem with global impact. Therefore, it is essential to have solid and well-documented economic evaluations that enable the prioritization of interventions and the efficient allocation of resources, as well as the design of programs that mitigate the health and economic impacts. The paper rigorously analyzes the available literature on economic evaluations, clearly describing the search strategy, selection, and critical analysis of the included studies. I have only a few observations.
Response 1: Thank you very much. Please find our responses to your comments below.
Comment 2: First, it is recommended that Figure 2 be reviewed and corrected, as it appears incomplete in the current document, making it difficult to correctly interpret the information it seeks to present.
Response 2: Thank you. Figure 2 in the previously shared document was not properly aligned within the page margins. We have now resized it to fit correctly.
Comment 3: On line 233, the country Columbia is mentioned; however, the correct name is Colombia. The same is true for supplementary materials 6 and 7.
Response 3: Thank you. This has now been corrected.
Comment 4: It is also recommended to write the name of the microorganisms in italics throughout the document, especially in Table 3, with the first letter of the first name capitalized, for example: Staphylococcus aureus.
Response 4: Thank you. The correction has been applied consistently throughout the manuscript.
Reviewer 2 Report
Comments and Suggestions for Authors
Dear authors,
The manuscript "Existing Evidence from Economic Evaluations of Antimicrobial Resistance – A Systematic Literature Review" constitutes a relevant contribution to the reflection on this current topic. It has become a reference for science and technology.
The methodology used for the systematic review is the PRISMA model. The inclusion/exclusion criteria and the content are demonstrated. Eight supplements are included to support the review and methodology.
The robustness of the available evidence from economic evaluations of AMR is demonstrated in Table 3. Table 4 highlights the associated limitations.
The authors discuss the results, presenting the limitations and trends identified, with the strong reflection/conclusion that there are still persistent evidence gaps and recurring methodological shortcomings in antimicrobial resistance economic evaluations.
A correction is necessary in Figure 2, as it is "cut."
Author Response
Comment 1: The manuscript "Existing Evidence from Economic Evaluations of Antimicrobial Resistance – A Systematic Literature Review" constitutes a relevant contribution to the reflection on this current topic. It has become a reference for science and technology. The methodology used for the systematic review is the PRISMA model. The inclusion/exclusion criteria and the content are demonstrated. Eight supplements are included to support the review and methodology. The robustness of the available evidence from economic evaluations of AMR is demonstrated in Table 3. Table 4 highlights the associated limitations.
The authors discuss the results, presenting the limitations and trends identified, with the strong reflection/conclusion that there are still persistent evidence gaps and recurring methodological shortcomings in antimicrobial resistance economic evaluations.
Response 1: Thank you.
Comment 2: A correction is necessary in Figure 2, as it is "cut."
Response 1: Thank you. The figure was not properly aligned within the document margins; we have now corrected it.
Reviewer 3 Report
Comments and Suggestions for Authors
The manuscript “Existing Evidence from Economic Evaluations of Antimicrobial Resistance – A Systematic Literature Review” Gunarathna et al. is clear, well-written, and very useful to the community of institutions, such as WHO and governments, as well as to researchers focused on AMR or the economic cost of diseases. This manuscript represents a lot of work, and I congratulate the authors on it.
I have two suggestions:
- I would prefer to move the big tables 3, 4, 5, and 6, to an Appendix (but not to the Suppl. Files because these tables should stay in the manuscript) instead of putting them in the middle of the text. I think it would increase the readability of the paper.
- In lines 341-342 I think the sentence “Moreover, a study conducted across 139 hospitals in seven Eastern Mediterranean countries revealed high overall hospital antimicrobial usage in the Eastern Mediterranean region” should be simplified to, e.g., “Moreover, a study conducted across 139 hospitals in seven Eastern Mediterranean countries revealed high overall hospital antimicrobial usage”.
Author Response
Comment 1: The manuscript “Existing Evidence from Economic Evaluations of Antimicrobial Resistance – A Systematic Literature Review” Gunarathna et al. is clear, well-written, and very useful to the community of institutions, such as WHO and governments, as well as to researchers focused on AMR or the economic cost of diseases. This manuscript represents a lot of work, and I congratulate the authors on it. I have two suggestions:
Response 1: Thank you. We have provided responses to each of your comments.
Comment 2: I would prefer to move the big tables 3, 4, 5, and 6, to an Appendix (but not to the Suppl. Files because these tables should stay in the manuscript) instead of putting them in the middle of the text. I think it would increase the readability of the paper.
Response 2: Thank you. We understand that including large tables may affect the readability of the paper. Therefore, we have moved the largest table (Table 3) to the appendix (Appendix 1). However, we did not move Tables 4–6, as we believe they are essential to be presented in their respective sections of the manuscript. To optimize space, we have reformatted Tables 4–6. We also anticipate that the space occupied by these tables may be further reduced during the article production stage if the manuscript is accepted for publication.
Comment 3: In lines 341-342 I think the sentence “Moreover, a study conducted across 139 hospitals in seven Eastern Mediterranean countries revealed high overall hospital antimicrobial usage in the Eastern Mediterranean region” should be simplified to, e.g., “Moreover, a study conducted across 139 hospitals in seven Eastern Mediterranean countries revealed high overall hospital antimicrobial usage”.
Response 3: Thank you. This has been corrected.